# Experimental Study on the Sulfuric Acid Corrosion Resistance of PHC Used for Pipe Pile and NSC Used in Engineering

Jie Xiao [1], Huanqiang Huang [1], Hehui Zeng [1], Lingfei Liu [2], Long Li [3], Haibo Jiang [1,*], Zucai Zhong [1] and Anyang Chen [1,4]

[1] School of Civil and Transportation Engineering, Guangdong University of Technology, Guangzhou 510006, China; xiaojie2017@gdut.edu.cn (J.X.); 2112209117@mail2.gdut.edu.cn (H.H.); 2112209115@mail2.gdut.edu.cn (H.Z.); taojiangxiaojie@126.com (Z.Z.); aychen@stpt.edu.cn (A.C.)

[2] School of Transportation, Civil Engineering & Architecture, Foshan University, Foshan 528225, China; lingfeiliu@fosu.edu.cn

[3] Guangdong Sanhe Building Materials Group Co., Ltd., Zhongshan 528414, China; sanhepile@sanhepile.com

[4] Department of Construction and Ecology, Shantou Polytechnic, Shantou 515071, China

[*] Correspondence: hbjiang@gdut.edu.cn

**Abstract:** In order to compare and investigate the sulfuric acid corrosion resistance of concrete for PHC pipe piles and two kinds of concrete commonly used in engineering, acid accelerated corrosion tests were conducted on specimens with three different strength grades of C30, C50, and C80 in a sulfuric acid solution with pH $\approx$ 0.85. The appearance of the specimens was observed, and the changes in mass loss percentage, corrosion depth, and stress–stain curves under uniaxial compressive loading were calculated and obtained with the corrosion time. From the comparison of corrosion depth and mass loss percentage of the concrete specimens with three different strength grades of C30, C50, and C80, it was found that the higher the strength grade of the concrete, the more severe the corrosion degree. The shapes of the stress–strain curves of three different strength grades of concrete specimens were basically the same. As the corrosion time was prolonged, the peak stress and the elastic modulus of concrete decreased. From the perspective of long-term corrosion, C80 specimens had a relatively smaller percentage of peak stress loss and a stronger resistance to peak stress loss. The research results provide references for the durability design of concrete structures and the prediction of concrete's service life in a sulfuric acid environment.

**Keywords:** concrete corrosion; sulfuric acid attack; strength grade; corrosion depth; mass loss; mechanical properties

## 1. Introduction

Under normal environmental conditions, the advantages of high strength, low cost, convenient materials, easy construction and molding, durability, and so on make concrete the most widely used building material in foundation construction [1–4]. The prestressed high-strength concrete (PHC) pipe pile is a kind of precast concrete component with a hollow cylinder body made by steam curing by prestressing centrifugal forming process, and its concrete strength grade is above C80 [5]. The PHC pipe pile has been widely used in industrial and civil buildings, wharfs, ports, and long-span bridges, and other projects, because it has many advantages such as a short construction period, strong quality control, a high bearing capacity, a low cost, fast construction, and so on [5–11]. However, concrete is a heterogeneous material, and there are also some inherent weaknesses, such as low tensile strength, high brittleness, poor acid resistance, and poor permeability. Since the cement hydration products are alkaline, the pH value of concrete generally ranges between 12 and 13.5 [12], so it erodes easily in an acidic environment [13–15] and its good durability is greatly weakened.

In urban and industrial production areas, sulfuric acid environments are widespread. Sulfur oxides and nitrogen oxides produced by fossil fuel combustion and vehicle exhaust are oxidized into sulfuric acid and nitric acid in the atmosphere, and these acidic substances fall with rain and snow, forming acid rain [16]. At present, sulfuric acid rain exists widely in the world [17]. In the concrete sewage pipeline system, the sulfur-containing organic matter in the sewage is converted into hydrogen sulfide under the action of bacteria, and the generated hydrogen sulfide is absorbed into the wet upper surface of the concrete pipeline, and sulfuric acid and other sulfur-containing by-products are formed after a series of reactions; the generated sulfuric acid is the main cause of corrosion in concrete sewage pipelines [18–20]. When concrete tunnels pass through pyrite-rich soil strata, sulfuric acid is produced by the oxidation of pyrite, which causes the performance degradation of the concrete tunnel lining [21]. Although concrete structures are generally designed for a minimum period of 50 years, their lifespan is sometimes shortened to a few years due to sulfuric acid attacks [22], which may cause a great waste of resources. The sulfuric acid first undergoes an acid-base neutralization reaction with the calcium hydroxide in the concrete to produce gypsum. Subsequently, gypsum reacts with calcium aluminate hydrate to form ettringite. The volume of these two corrosion products will be larger than the volume represented by the initial compound, leading to concrete cracking [23]. Therefore, the durability of concrete materials and structures in a sulfuric acid environment has attracted extensive attention from experts and scholars at home and abroad [24–27].

The corrosion effect of sulfuric acid on concrete materials is a very complicated process, and is affected by many factors, which can be divided into internal factors such as the type of cement, the content of the cement, the supplementary cementing materials, and the aggregate, and external factors, such as the pH value of the sulfuric acid solution, the fluidity of solution, the stress state of the concrete, and the temperature. However, there is currently no standardized procedure to test concrete's resistance to sulfuric acid attack [28]. Because of a lack of standardized methods, different test methods have been used and various parameters have been modified to evaluate the resistance of the materials [19]. Živica V. [29] described four types of index parameters to characterize the performance of concrete or mortar in an acidic environment: 1. Appearance, such as length and corrosion depth; 2. Mechanical performance indicators, such as mass, strength, and dynamic elastic modulus changes; 3. Chemical composition, such as the change in calcium ion concentration in the solution and the change in the pH value of the solution; and 4. The microscopic pore structure. Fattuhi N. I. et al. [30] studied the influence of the water–cement ratio (or cement content) on the corrosion of cement-based materials in sulfuric acid solution, and found that with the increase in the water–cement ratio, the mass loss of specimens decreases, because concrete specimens with a low water–cement ratio have a high cement content and are more susceptible to corrosion by sulfuric acid. Israel D. et al. [31] investigated the corrosion resistance of pore-reduced cement (PRC) and ordinary Portland cement (OPC) in corrosive media such as sulfuric acid, hydrochloric acid and ethanoic acid by regular visual inspection and sample weighing. It was found that in the case of hydrochloric acid and ethanoic acid attack, PRC was more resistant to corrosion than OPC, because the closer microstructure had better resistance to the attacks of hydrochloric acid and acetic acid. Sulfuric acid damage to PRC and OPC was almost the same, because sulfuric acid attacks occur primarily on the outer surface, and the dense microstructure of PRC is not beneficial.

In order to simulate the degradation of the sewage system and manure storage structures, De Belie N. et al. [32] adopted a sulfuric acid solution with a concentration of 0.5% (pH = 0.8~1.0) to carry out an accelerated corrosion test on a concrete cylindrical core, measured the change in radius of the concrete cylinder by a laser sensor, and calculated the evolution of the surface roughness of the concrete cylinder. Chang Z. T. et al. [33] immersed six kinds of concrete cylinder specimens in 1% sulfuric acid solution for 168 days, and periodically examined for surface appearance deterioration, mass loss and compressive bearing capacity. The test results showed that the concrete containing limestone aggregate, silica fume, and fly ash had the best performance because the limestone aggregate could be used

as a sacrificial medium to reduce the acid concentration near the surface of the concrete. In addition, they believed that the compressive bearing capacity of the concrete cylinder was a more reliable performance indicator to evaluate the deterioration of concrete under a sulfuric acid environment than the mass loss. Muthu M. et al. [34] soaked the graphene oxide (GO)-modified cement slurry in 1 M sulfuric acid for two weeks and found that the mass and cross-sectional area loss of the sample containing GO was low during sulfuric acid exposure and improved as the GO content increased. This was attributed to the fact that the addition of GO makes the microstructure of the concrete denser, thus limiting the entry of erosive ions from the external solution into the interior of the concrete. Cao R. et al. [35] comparatively analyzed the sulfuric acid resistance of an Ordinary Portland cement (OPC)/Calcium Sulfoaluminate Cement (CSA) composite system, and evaluated the deterioration from the aspects of visual appearance, mass change, compressive strength loss, and microstructure. The results showed that the OPC-CSA composite material has the potential to improve the corrosion resistance of the cement matrix to sulfuric acid. Previous studies have focused on the effects of aggregate type, pH value and mineral admixtures on the sulfuric acid resistance of concrete, and relatively little research has been conducted on the sulfuric acid corrosion resistance of concrete for the PHC pipe pile and concrete commonly used in engineering.

The pre-stressed high-strength concrete (PHC) pipe pile is in the groundwater for a long time, or in an alternating wet and dry environment for a long time, and compared with the superstructure members the PHC pipe pile is in a more severe environment. Moreover, the PHC pipe pile is usually buried underground, so it is difficult to carry out a visibility inspection, and it is difficult to find corrosion if it occurs. The concealment of PHC pipe piles makes it easy for them to be neglected, and more attention should be paid to the durability design of PHC pipe piles [36–38]. Based on this, in this paper, concrete samples with three different grade strengths of C30, C50 and C80 were selected, and a sulfuric acid solution with a pH of 0.85 was used as the corrosive medium to conduct an accelerated corrosion test on concrete. The time-varying law of appearance, mass loss, corrosion depth, and mechanical properties under the uniaxial compression of PHC pipe pile concrete (represented with C80) and two kinds of concrete commonly used in engineering (represented with C30 and C50) samples in the sulfuric acid environment were studied. The research results will provide references for the durability design of concrete structures and the prediction of concrete's service life in a sulfuric acid environment.

## 2. Experiment Design and Test Methods

### 2.1. Experiment Design

Local Portland cement (42.5 P II) produced by China Resources Cement Holdings Limited was used, which complied with Chinese standard GB175-2007. Granite gravel was used as a coarse aggregate and machine-made sand was used as the fine aggregate. In order to represent the concrete material used for PHC pipe piles, the mix ratio of C80 concrete specimens was consistent with that of a PHC pipe pile plant in Guangdong province, where the forming and curing of specimens were also carried out. The C50 and C30 concrete specimens were poured at a mixing plant near the PHC pipe plant in order to represent the concrete commonly used in engineering. Mixture proportions for concrete specimens are given in Table 1. Naphthalene-based superplasticizer was added to obtain sufficient workability. The production of pipe piles is mainly achieved through high-speed centrifugal molding of molds, which generates a large amount of residual slurry during the molding process. In addition to water, the main components of the residual slurry produced by pipe pile production are cement, ground sand, a small amount of fine sand, and very little water reducing agent. The residual slurry contains about 70% liquid and 30% solid. After years of testing, the pipe pile enterprise has obtained a surplus slurry content that meets the performance requirements, and has achieved good economic and environmental protection benefits. The specific surface area of the ground sand used is greater than 420 m$^2$/kg, and the silicon dioxide content is greater than 90%. When the ground sand is added to

cement concrete, it can further react with the silicon dioxide in the ground sand and the calcium hydroxide in the hydration product of cement under autoclave curing conditions to generate tobermorite, with high strength, good crystallinity, and stability.

**Table 1.** Mixture proportions of C30, C50, and C80 concrete (kg/m$^3$).

| Strength Grade | Cement | Fly Ash | Mineral Powder | Ground Sand | Sand | Gravel | Superplasticizer | Residual Slurry | Water |
|---|---|---|---|---|---|---|---|---|---|
| C30 | 198 | 66 | 66 | / | 780 | 1075 | 10.8 | / | 155.0 |
| C50 | 255 | / | / | 135 | 750 | 1300 | 9.5 | 150 | / |
| C80 | 255 | / | / | 135 | 720 | 1330 | 9.5 | 180 | / |

Granite gravel with a maximum nominal size of 31 mm was obtained for the concrete specimens. Table 2 shows the particle size gradation of gravel (coarse aggregates) used in the trial mixture.

**Table 2.** Results of the sieving test for granite gravel.

| Sieve size (mm) | 31.5 | 26.5 | 19.0 | 16.0 | 9.5 | 4.75 | 2.36 | <2.36 |
|---|---|---|---|---|---|---|---|---|
| Grader retained (%) | 0 | 1.5 | 43.5 | 6.4 | 36.4 | 2.7 | 4.1 | 5.4 |
| Accumulated retained (%) | 0 | 2 | 45 | 51 | 88 | 91 | 95 | 100 |

Specimens C30, C50, and C80 were all cylinders with diameters of 100 mm and with a height of 200 mm. In order to accelerate the corrosion process, all specimens were immersed in a sulfuric acid solution with pH ≈ 0.85 after curing. The test design scheme is shown in Table 3.

**Table 3.** Grouping of specimens.

| Specimen Type | Concrete Grade Strength | pH Value | Quantity/Piece | Immersion Method |
|---|---|---|---|---|
| Cylinders | C30 | 0.85 | 4 group × 3 = 12 | Full immersion |
| Cylinders | C50 | 0.85 | 4 group × 3 = 12 | Full immersion |
| Cylinders | C80 | 0.85 | 4 group × 3 = 12 | Full immersion |

Note: In order to fully consider the discreteness of the test results, 3 specimens were selected for each test group, and the test results were the average value of the test results of the group.

### 2.2. Test Methods

According to the corrosion time (in this paper, set as 0 days, 28 days, 56 days, and 165 days), the test specimen was completely immersed in a pH ≈ 0.85 sulfuric acid solution. A portable pH meter (0.01 precision) was used to measure the pH of solutions during the testing period of 165 days. It was kept in the range of 0.83–0.87 by adjusting the pH value daily using concentrated sulfuric acid (98% by weight). The solution was thoroughly stirred after adding the concentrated sulfuric acid every day in order to reduce differential concentrations of the acid within the solution vessel. After the immersion was completed, it was removed regularly from the sulfuric acid solution and dried in the oven. The mass of each specimen before and after corrosion was measured using an electronic scale with an accuracy of 1 g and a range of 30 kg. The mass loss of the cylindrical specimen was calculated according to the following equation:

$$K_{mi} = (m_i - m_0)/m_0 \times 100\% \tag{1}$$

where $K_{mi}$ is the percentage of mass loss, %, and $m_0$ and $m_i$, respectively, represent the mass of the cylinder before corrosion and after i days of corrosion, kg.

The diameter measurement adopted a 0–300 mm digital display vernier caliper with a resolution of 0.01 mm. As shown in Figure 1, three equally spaced positions, A-A′, B-B′, and C-C′, were selected along the axis of the cylindrical specimen in the 1–2 direction,

and then the cylinder was rotated 90 degrees. Similarly, three positions were selected along the axial direction of the cylinder with equal spacing in the direction 3–4, and a total of 6 measurements were made. The average of the six diameter measurements of each specimen were taken as the diameter of the specimen. The corrosion depth of the cylindrical specimen was calculated according to the following equation:

$$h = (d_i - d_0)/2 \tag{2}$$

where h represents the corrosion depth, mm, and $d_0$ and $d_i$, respectively, represent the diameter of the cylinder before corrosion and after i days of corrosion, mm.

In order to ensure that the upper and lower end faces of the cylinder conformed with the planeness requirements of applicable standards, high-strength gypsum caps were formed against a glass plate. The cylinder specimen uniaxial compression test was conducted using a servo-hydraulic testing machine MATEST C088-10 produced in Italy with a compression load capacity of 4000 kN. Because the surface of the concrete specimen will generate loose and porous corrosion products after corrosion, the traditional strain gauge cannot be firmly attached to the surface of the specimen. Even if the surface of the specimen is polished and leveled, the accuracy of the data recorded by the strain gauge cannot be guaranteed. In this article, high-accuracy displacement meters were used to collect longitudinal strain data of the specimens and draw the full stress–strain curve of the specimens under compression. Two high-accuracy linear variable differential transducers (LVDT) were symmetrically fixed on both sides of the cylindrical specimen using self-made clamping fixtures to measure the average axial compressive strain of 80 mm in the middle section of the cylinder, as shown in Figure 2. The cylindrical specimens were firstly preloaded by force-controlled loading until the force stabilized at 10 kN to ensure the normal operation of the instrument and to ensure that there was no obvious eccentric compression on the specimen. Then, the specimens were formally loaded by displacement-controlled loading with a displacement rate of 0.12 mm/min. In the loading process, the applied load, displacement, and strain data were collected with JM3813 multifunctional static strainometer every 1 s, until the test stopped when the load dropped 80% of the peak load. Sufficient data would then be obtained to establish a complete stress–strain curve, including ascending and descending branches.

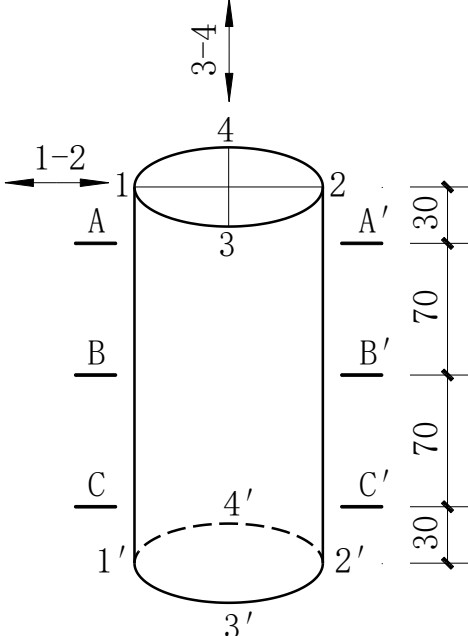

(**a**) Diagram of the diameter measurement position.

**Figure 1.** *Cont.*

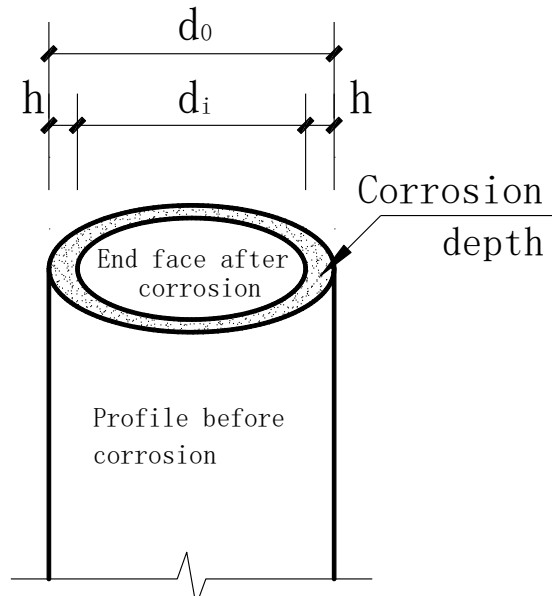

(**b**) Schematic diagram of corrosion depth calculation.

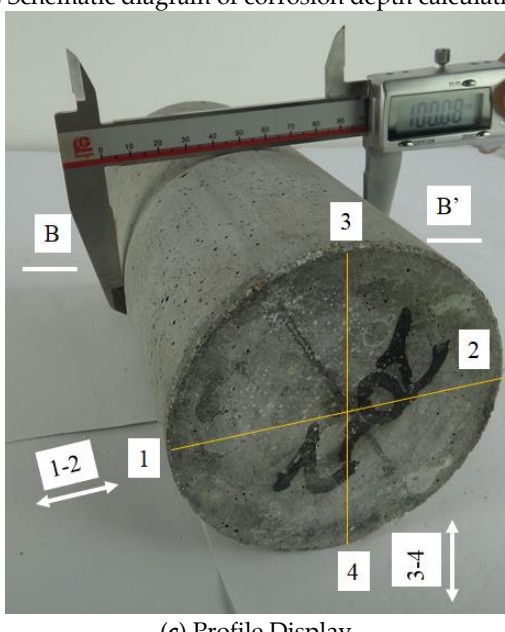

(**c**) Profile Display

**Figure 1.** Schematic diagram of cylinder dimension measurement and corrosion depth.

According to the crushing load measured in the experiment and the area of the cylindrical specimen obtained from the vernier caliper test, the peak stress of the concrete specimen before and after corrosion was calculated as in Equation (3). The peak stress loss of the cylindrical specimen was calculated according to Equation (4).

$$f_0 = F_0/A_0 \, , f_i = F_i/A_i \tag{3}$$

$$K_{fi} = (f_i - f_0)/f_0 \times 100\% \tag{4}$$

where $f_0$, $F_0$, and $A_0$ represent the average peak stress, crushing load, and area of a group of three specimens before corrosion. $f_i$, $F_i$, and $A_i$ represent the average peak stress, crushing load, and area of a group of three specimens after i days of corrosion. $K_{fi}$ represents the peak stress loss of the cylindrical specimen.

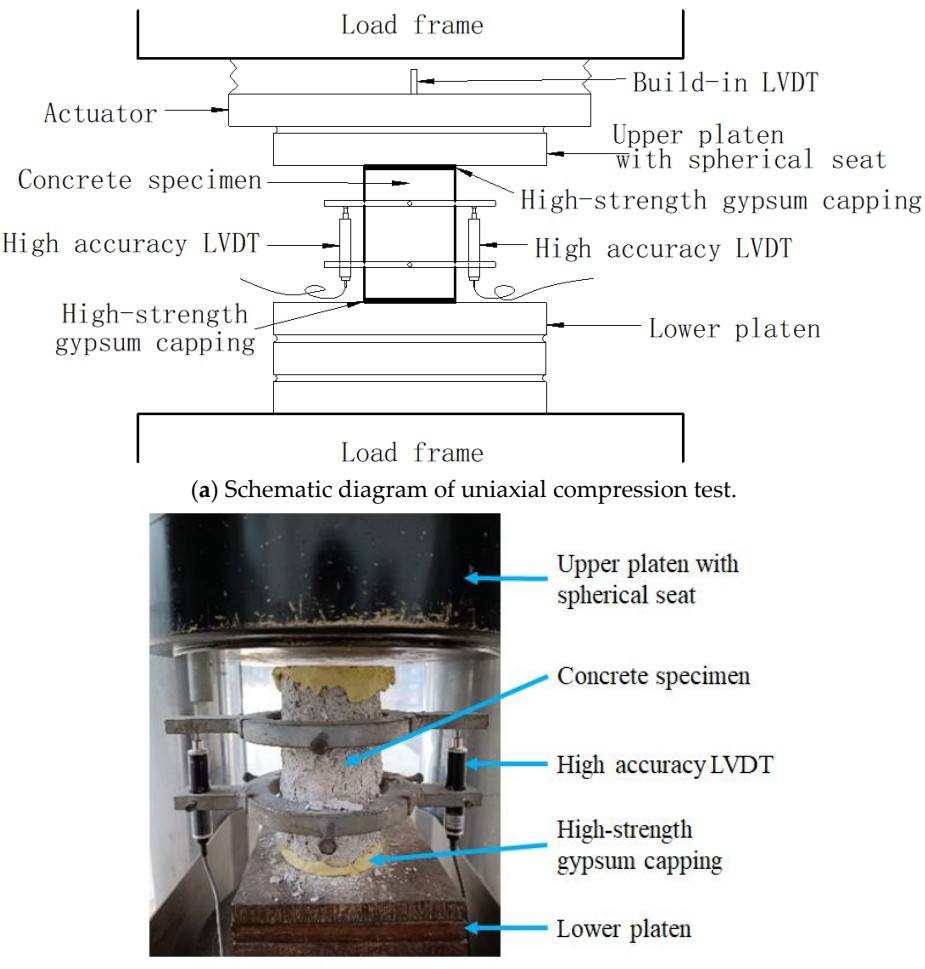

(**a**) Schematic diagram of uniaxial compression test.

(**b**) Profile Display

**Figure 2.** Schematic diagram of a uniaxial compression servo-hydraulic testing machine for cylindrical specimens.

## 3. Experiment Results and Analysis

### 3.1. Visual Appearance

The change in the appearance and morphology of concrete cylinder specimens can directly reflect the degree of sulfuric acid corrosion. Photographs were taken of cylinder samples after immersion in the sulfuric acid over different periods to record changes in the surface appearance. Figure 3a,b show the visual appearance of the concrete specimens before corrosion and after 165 days of sulfuric acid exposure. Slight corrosion occurred on the surface of the C30 concrete specimen, which was characterized by the color of the specimen changing from stone gray to grey-white, and little cement pastes being dissolved. However, after observing the appearance of the C80 concrete specimen after 165 days of corrosion, it was found that the concrete specimens suffered structural damage, and the coarse and fine aggregates on the surface were exposed after the dissolution of the cement paste. A layer of loose and porous surface structures had plenty of corrosion products attached, and some corrosion products had significantly peeled off, making the surface very rough and uneven. A layer of loose porous corrosion products was also observed on the surface of the C50 concrete specimen. However, compared with the C80 concrete specimen, the corrosion products of the C50 concrete specimen had less spalling, the surface roughness was smaller, and the coarse aggregate was less exposed. From the surface observation, C80 concrete specimens suffered the most severe corrosion, followed by the C50 concrete specimens, and the C30 concrete specimens suffered the least severe corrosion.

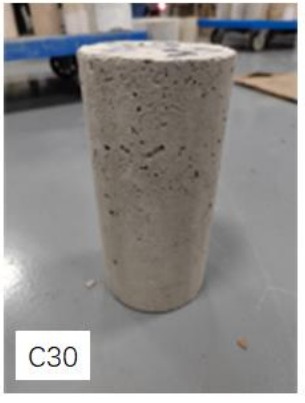 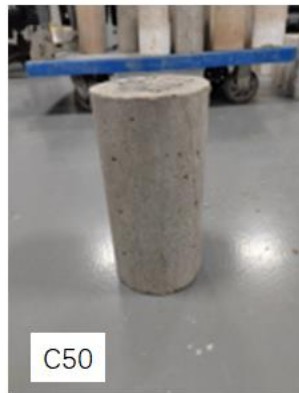 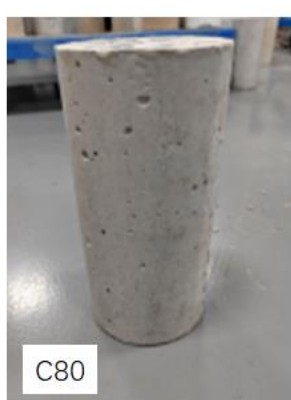

(**a**) Visual appearance of C30, C50, and C80 before corrosion.

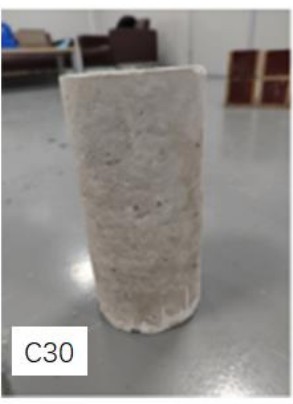 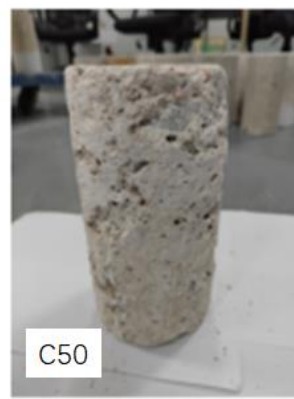 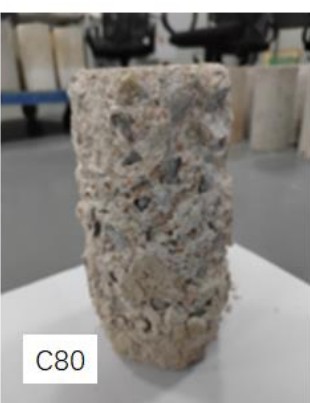

(**b**) Visual appearance of C30, C50, and C80 after 165 days of corrosion.

**Figure 3.** Comparison of the visual appearance of C30, C50, and C80 concrete specimens before and after sulfuric acid corrosion.

### 3.2. Corrosion Depth

The relationship between the corrosion depths of concrete with three strength grades of C30, C50, and C80 and the corrosion time is plotted in Figure 4. It should be noted that although it is easier to see the trend of changes by connecting scattered points into a line, it is not scientific enough because we only obtained data at the scattered points in our experiment. Therefore, in this article, we used the method of drawing scattered points to reflect the experimental results. It can be seen from Figure 4 that in the early stage of corrosion, the corrosion depths of all three strength grades of C30, C50, and C80 of concrete showed an increasing trend. As the corrosion time continued to increase, the diameter of C30 continued to increase, while the diameters of C50 and C80 decreased. This is because in the early stages of corrosion, the concrete reacts with sulfuric acid and expansive white corrosion products are deposited on the surface of the specimen, leading to an increase in the diameter of the specimen and an increase in the corrosion depth. However, with the increase in corrosion time, corrosion products are generated from the surface to the inside. The specimen will be further damaged and the surface corrosion layer will fall off, resulting in a smaller diameter of the specimen and a larger corrosion depth. Moreover, the more severe the corrosion is, the more severe the detachment that occurs, and the smaller the diameter becomes. Therefore, based on the corrosion depth, the severity of corrosion could be ranked as C80 > C50 > C30. This is because the higher the strength grade, the smaller the water–cement ratio, and the more alkaline hydration products in the unit volume, which then react with sulfuric acid to generate and shed more corrosion products.

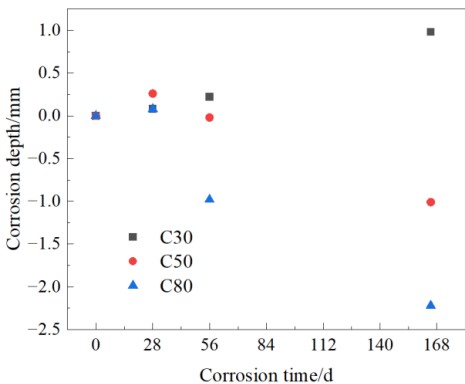

**Figure 4.** Corrosion depth changes with corrosion time.

### 3.3. Mass Loss

The variation curve of the specimen mass loss with the corrosion time for three strength grades of C30, C50, and C80 of concrete under sulfuric acid solution corrosion is shown in Figure 5. As can be seen from Figure 5, the masses of C30, C50, and C80 concrete specimens all increased rapidly after 28 days of sulfuric acid immersion. Because in the early stage of corrosion the alkaline substances in concrete underwent a faster neutralization reaction with sulfuric acid, resulting in more corrosion products, and there were many products generated by corrosion, so the quality increased at the macro level. When the immersion time was between 28 and 56 days, the mass of C30, C50, and C80 concrete specimens continued to increase, but the rate of increase was slower than the previous period. At this stage, the reason the mass loss rate slowed down was that corrosion products continuously deposit and adhere to the surface of the concrete specimens, which hinders the penetration of corrosive media and makes the slope of the curve of the relationship between the mass loss rate and the corrosion time smaller. When the immersion time was between 56 and 165 days, the mass of the C30 concrete specimen continued to increase, while the masses of C50 and C80 both showed a downward trend. However, the mass of the C50 concrete specimen was always higher than the initial mass, while that of the C80 concrete specimen rapidly decreased, and the mass loss percentage of the C80 concrete specimen reached −5.07% after 165 days of sulfuric acid immersion. The reason that the mass of the C30 concrete specimens was still increasing in the later period was that the porosity of the C30 specimens was relatively large, and the generated corrosion products did not accumulate enough for cracking and spalling concrete. However, the massed of C50 and C80 specimens decreased because when the corrosion products accumulate to a certain extent, the expansion of corrosion products will cause the concrete to crack and damage, and finally the corroded concrete will peel off and result in a mass decrease. From the curves comparison in Figure 5, the higher the strength grade, the greater the specimen mass change.

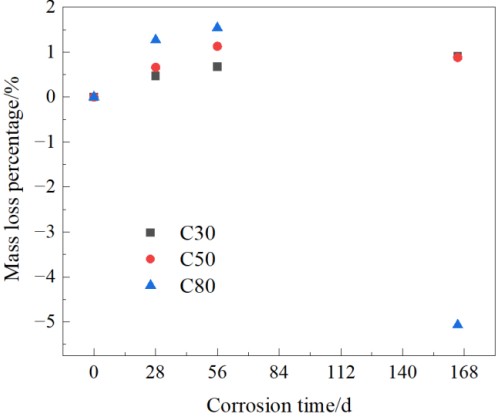

**Figure 5.** The percentage of mass loss changes with corrosion time.

### 3.4. Experimental Stress–Strain Curves

Under axial compressive load, the stress–strain curve for three strength grades of C30, C50, and C80 of concrete under sulfuric acid solution corrosion is shown in Figure 6, and we selected the curve whose strength was the middle value in a group of three specimens. As can be seen from Figure 6, the stress–strain curve of concrete under different corrosion ages did not change significantly, and its shape was basically the same. As shown in Figure 6, it can be seen that as the corrosion time prolonged from 0 days to 165 days, the stress–strain curve of concrete tended to flatten, and the initial slope of the curve decreased. The slope of the ascending branches of the curve decreases, while the descending branch of the stress–strain curve tended to be gentle, indicating a decrease in the elastic modulus of concrete and an increase in the ultimate strain compared to uncorroded concrete. Another characteristic difference between the stress–strain curves of corroded concrete and uncorroded concrete was that the peak points shifted downwards and backwards, indicating that as the corrosion time prolongs, the peak stress of concrete decreases, the peak strain gradually increases, and the concrete corroded by sulfuric acid shows a gradual loosening trend.

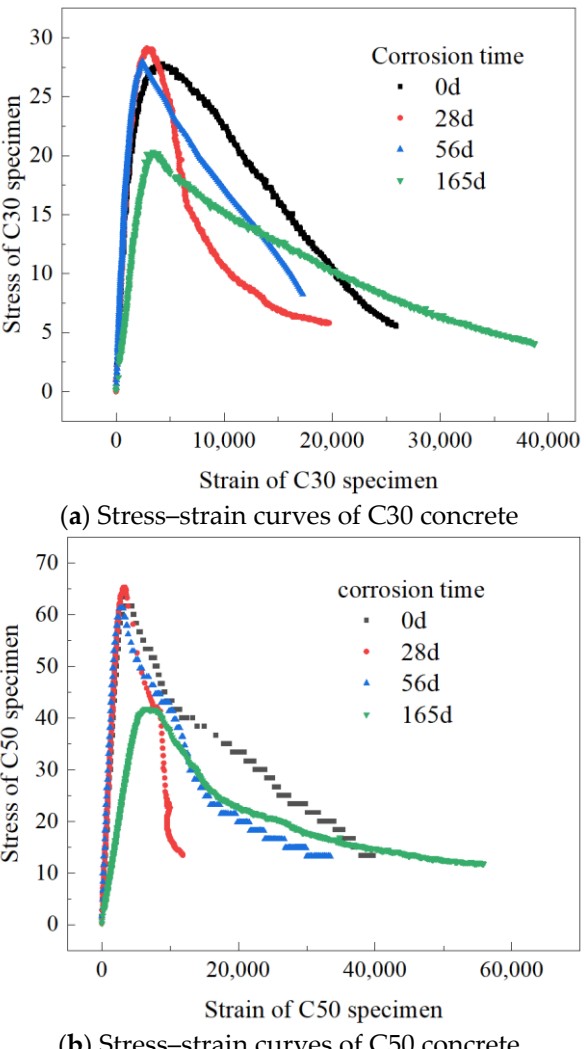

(**a**) Stress–strain curves of C30 concrete

(**b**) Stress–strain curves of C50 concrete

**Figure 6.** *Cont.*

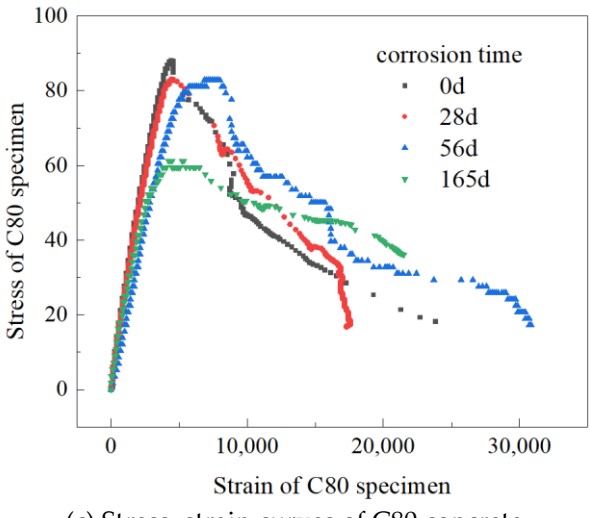

(**c**) Stress–strain curves of C80 concrete

**Figure 6.** Stress–strain curves of sulfuric-acid-corroded concrete under axial compressive load.

*3.5. Peak Stress and Elastic Modulus*

The peak stress curve of C30, C50, and C80 concrete specimens corroded by sulfuric acid solutions with corrosion time is shown in Figure 7. As can be seen from Figure 7, the peak stress of concrete specimens with three strength grades of C30, C50, and C80 was basically consistent with the change trend in corrosion time. In the early stage of corrosion, the peak stress changed gently, but with the increase in corrosion time, the peak stress showed an obvious downward trend after 28 days of immersion in sulfuric acid. These phenomena may be caused by the gradual formation of corrosion products in the early stage, which increases the compactness of the concrete and thus increases its strength. However, with the increase in corrosion time, the excessive accumulation of corrosion products leads to the increase in micro-cracks in the concrete, leading to the deterioration of the strength of the concrete. Kawai [39] found that concrete with a high water cement ratio has larger and more pores than that with a low water–cement ratio. These pores play the role of a capacity to absorb expansion caused by the production of gypsum. Therefore, concrete with a high water–cement ratio has a higher capacity to absorb the expansion of the production reaction of gypsum than that with a low water–cement ratio, and this phenomenon is schematically illustrated in Figure 8.

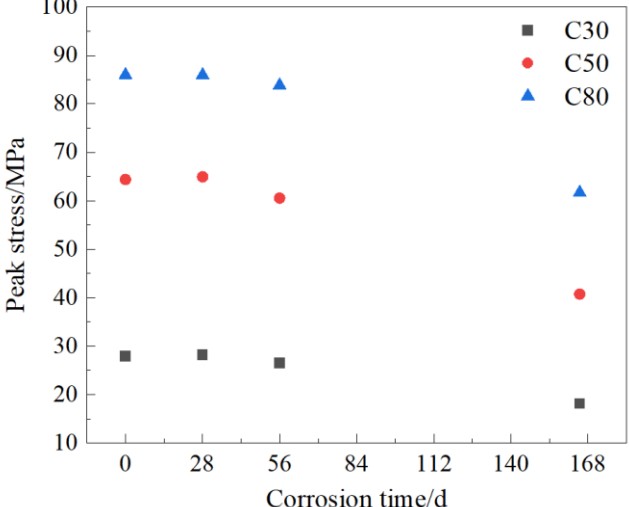

**Figure 7.** Variation of peak stress with corrosion time.

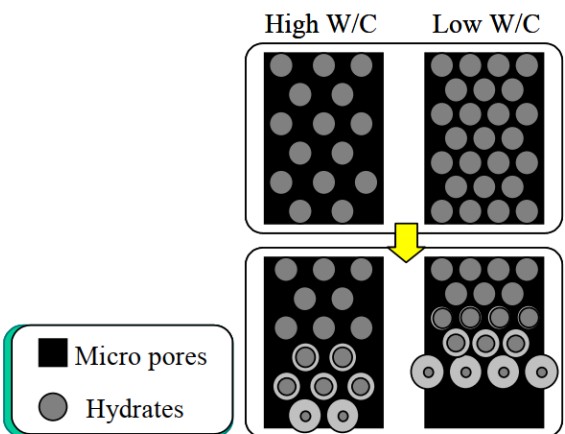

**Figure 8.** Mechanism of concrete deterioration due to sulfuric acid attack.

The percentage of peak stress loss varies with the corrosion time, as shown in Figure 9. Both C30 and C50 specimens showed a trend of slowly increasing at first and then rapidly decreasing. After 165 days of accelerated corrosion in sulfuric acid solution, the peak stress of C30, C50, and C80 specimens decreased by 34.97%, 36.76%, and 28.27%, respectively. From the perspective of long-term corrosion, C80 specimens had a relatively smaller percentage of peak stress loss and a stronger resistance to peak stress loss. This is because C80 had a lower water–cement ratio, lower porosity, and a relatively denser interior. However, the resistance of C30 and C50 specimens to peak stress loss was weak and similar.

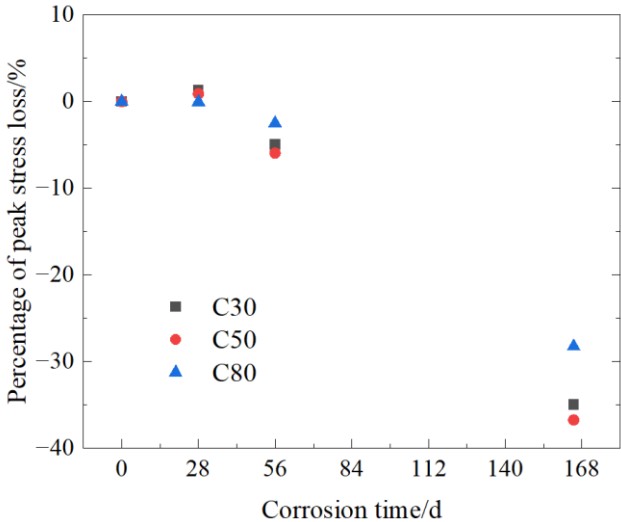

**Figure 9.** Variation rate of compressive strength with corrosion age.

Liu [40] drew a schematic diagram of the comparison of concrete before and after corrosion in a sulfate environment based on a large number of corrosion test results, as shown in Figure 10. Due to the diffusion effect of corrosive media, concrete specimens undergo corrosion from the surface to the inside. The light-colored area in Figure 10b represents the uncorroded concrete, which is called the uncorroded area, while the dark-colored area represents the concrete corroded by corrosive media, which is called the corroded area. The degradation of concrete mechanical properties is mainly caused by the degradation of concrete mechanical properties in the dark corrosion zone.

The elastic modulus of the tested concrete cylinder specimens can be obtained from the stress–strain curve. In this paper, the elastic modulus of the tested concrete cylinder

specimens was determined as the secant modulus of its stress–strain curve at 40% peak stress [41–43], as shown in Equation (5).

$$E_c = \frac{\sigma_{0.4} - \sigma_0}{\varepsilon_{0.4} - \varepsilon_0} \qquad (5)$$

where $E_c$ represents the elastic modulus (MPa); $\sigma_0$ and $\sigma_{0.4}$ are the stresses corresponding to 0.5 MPa and 40% of the peak stress, respectively; and $\varepsilon_{0.4}$ and $\varepsilon_0$ are the strain at the stress level $\sigma_{0.4}$ and $\sigma_0$, respectively. The elastic modulus loss of the cylindrical specimen was calculated according to the following equation:

$$K_{Ei} = (E_i - E_0)/E_0 \times 100\% \qquad (6)$$

where $K_{Ei}$ represents the elastic modulus loss of the cylindrical specimen, $E_0$ represents the average elastic modulus of a group of three specimens before corrosion, and $E_i$ represents the average elastic modulus of a group of three specimens after i days of corrosion. The percentage of elastic modulus loss varied with the corrosion time, as shown in Figure 11. As shown in Figure 11, it can be seen that the static elastic modulus of C30 significantly increased before 56 d, while C50 and C80 both decreased. Ultimately, after 165 d of sulfuric acid corrosion, all three strength grades of C30, C50, and C80 decreased by nearly half, which indicates that sulfuric acid corrosion causes significant damage to the elastic stage of concrete. Comparing the static elastic modulus of the three strength grades of C30, C50, and C80, it was found that C80 had a relatively better ability to resist the loss of elastic modulus.

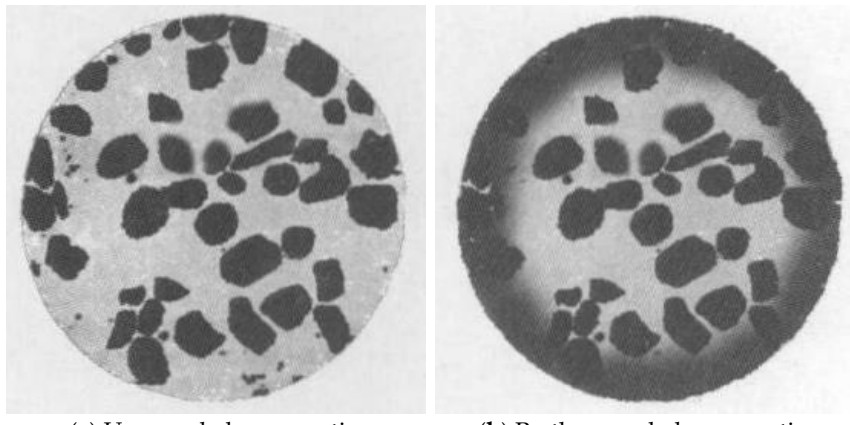

(**a**) Uncorroded cross section      (**b**) Partly corroded cross section

**Figure 10.** Schematic diagrams of the comparison of concrete before and after corrosion.

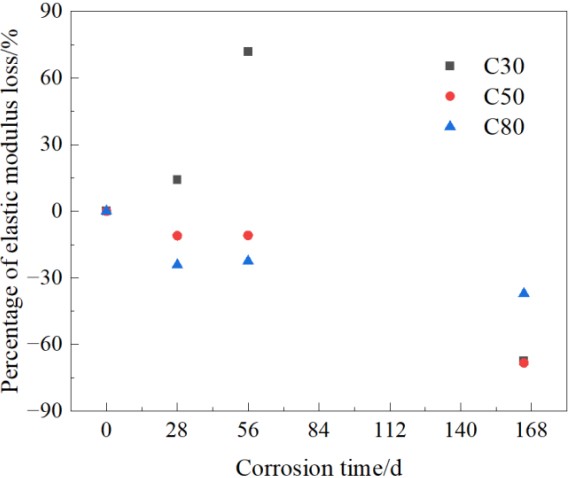

**Figure 11.** Variation in elastic modulus with corrosion age.

## 4. Conclusions

In this paper, PHC pipe pile concrete (represented by C80) and two kinds of conventional concrete (represented by C30 and C50) specimens were adopted to conduct accelerated corrosion tests in a sulfuric acid solution with a pH of 0.85. The time-varying law of appearance, mass loss, corrosion depth, and stress–stain curves under the uniaxial compressive loading of the specimens were explored. From this investigation, the following conclusions can be drawn.

After immersion in sulfuric acid solution with a pH of 0.85 for 165 days, white corrosion products were generated on the surface of concrete specimens of all three different strength grades of C30, C50, and C80. From the surface observation, C80 concrete specimens suffered the most severe corrosion, followed by C50 concrete specimens, and C30 concrete specimens suffered the least severe corrosion.

As the corrosion time continued to increase, the diameter and the mass of the C30 concrete specimens continued to increase. The reason was that the porosity of C30 concrete specimens was relatively large, and the generated corrosion products were deposited on the surface and pores of the specimens, but the corrosion products had not yet accumulated to the extent of concrete cracking and peeling.

During the period from 0 to 28 days, the diameter and mass of C50 and C80 specimens increased due to the adhesion of corrosion products on the surface of the specimens. From 28 to 165 days, the corrosion products on the surface of C50 and C80 specimens gradually fell off, resulting in a smaller diameter and larger corrosion depth. Between 28 and 165 days, the mass of C50 and C80 specimens increased slowly at first and then decreased, which was because the corrosion products attached to the surface hindered the penetration of the acidic media. When the corrosion products accumulated to a certain extent, the expansion of corrosion products caused concrete cracking and damage, and finally the corroded concrete would flake off and the mass would be reduced.

From the comparison of the corrosion depth and the mass loss percentage of concrete specimens with three different strength grades of C30, C50, and C80, the higher the strength grade of the concrete, the more serious the corrosion. This is because the higher the strength level, the smaller the water–cement ratio, the more alkaline hydration products in unit volume, and the more likely the reaction with sulfuric acid to form and shed more corrosion products.

Under axial compressive load, the stress–strain curve for three strength grades of C30, C50, and C80 of concrete under sulfuric acid solution corrosion did not change significantly, and its shape was basically the same. As the corrosion time was prolonged, the peak stress and the elastic modulus of the concrete decreased. After 168 days of accelerated corrosion in the sulfuric acid solution, the peak stress of C30, C50, and C80 specimens decreased by 34.97%, 36.76%, and 28.27%, respectively. From the perspective of long-term corrosion, C80 specimens had a relatively smaller percentage of peak stress loss and a stronger resistance to peak stress loss. This was because C80 had a lower water–cement ratio, lower porosity, and a relatively more dense interior. However, the resistance of C30 and C50 specimens to peak stress loss was weak and similar.

**Author Contributions:** J.X., Z.Z., L.L. (Lingfei Liu), L.L. (Long Li) and H.J. mostly contributed to the design of the manuscript. H.H. and H.Z. carried out data collection and processing. L.L. (Long Li), H.H., H.Z. and A.C. were involved in immersing the specimens. J.X., H.H., Z.Z., and L.L. (Lingfei Liu) wrote and revised the paper. J.X. and L.L. (Lingfei Liu) funding acquisition. All authors have read and agreed to the published version of the manuscript.

**Funding:** The authors appreciate the financial support provided by the National Natural Science Foundation of China, No. 52278160 and No. 51808133, the Innovation Project of Guangdong Graduate Education, with Grant No. 2019JGXM101, the Young Innovative Talents Project of Regular Universities in Guangdong Province, with Grant No. 2018KQNCX278, the Guangdong Natural Science Foundation, with Grant No. 2020A1515110814 and Grant No. 2023A1515012081, and the Science and Technology Planning Project of Guangzhou City, with Grant No. 202002030120.

**Data Availability Statement:** The data that support the findings of this study are available from the corresponding author upon reasonable request.

**Acknowledgments:** The authors would like to thank all the anonymous referees for their constructive comments and suggestions. The authors gratefully acknowledge the laboratory of the School of Civil and Transportation Engineering, Guangdong University of Technology, and Guangdong Sanhe Pipe Pile Co., Ltd. for providing the resources required for this study.

**Conflicts of Interest:** The authors declare no conflict of interest.

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
