# Peer review of "Experimental Study on the Sulfuric Acid Corrosion Resistance of PHC Used for Pipe Pile and NSC Used in Engineering"

_buildings, doi:10.3390/buildings13071596_

Round 1

Reviewer 1 Report

Dear Authors,

Thank you for your well written manuscript. Can you please match your abstract with conclusions and bring more values comparison in both as outcome of your study? It would be important to bring recommendations paragraph after discussions. Please pay attentions to your chapters (zero value shift) and possibility to read your charts independently from description in text. 

No comments

Author Response

Point 1: Dear Authors,

Thank you for your well written manuscript. Can you please match your abstract with conclusions and bring more values comparison in both as outcome of your study? It would be important to bring recommendations paragraph after discussions. Please pay attentions to your chapters (zero value shift) and possibility to read your charts independently from description in text.

Response 1: Thank you for your advice.

Reviewer 2 Report

The sulfuric acid corrosion resistance of PHC used for pipe pile and two normal strength concrete commonly used in engineering is experimentally studied in the present study. The topic of the study is interesting and significant. There are some issues should be addressed before the publication of the manuscript, and are listed below:

(1)     The title of the paper is suggested to be changed to “Experimental study on sulfuric acid corrosion resistance of PHC used for pipe pile and NSC commonly used in engineering”

(2)     The first paragraph of the introduction is too long, and it is recommended that it be divided into subparagraphs.

(3)     Some recently literatures are suggested to add and refer in the present paper. Such as " Permeability evolution model of coarse porous concrete under sulphuric acid corrosion. Construction and Building Materials, 326,2022, 126475; Corrosion of steel rebars across UHPC joint interface under chloride attack. Construction and Building Materials, 2023, 387: 131591; Experimental Study on Bond Behavior between Steel Rebar and PVA Fiber-Reinforced Concrete. Coatings, 2023, 13(4): 740. Corrosion-Effected Bond Behavior between PVA-Fiber Reinforced Concrete and Steel Rebar under Chloride Environment. Materials, 2023, 16(7): 02666;

(4)     The mixture proportion of C30, C50 and C80 designed in the present test should be added.

(5)     And, the material properties of the aggregates should also be added.

(6)     The size of each specimen in Figure 3 is recommended to be adjusted to the same.

(7)     The summary in lines 282-283 “the higher the strength grade, the greater the mass loss of the specimen” is inaccurate. As indicated that the sulfuric acid corrosion does not cause the mass loss in specimens with C30 and C50 in the present test.

(8)     Table 1 shows that 3 replicates were set up for each experimental condition, but only one curve and one peak stress are given in Figures 6 and 7 for each experimental condition, are they average values? How to give the average of the three load-slip curves? Please explain.

(9)     Lines 386-388 of the conclusion are an exact duplicate of lines 389-391.

(10)  Some quantitative results are suggested in lines 392-399 of the conclusion.

Author Response

Response to Reviewer 2 Comments

The sulfuric acid corrosion resistance of PHC used for pipe pile and two normal strength concrete commonly used in engineering is experimentally studied in the present study. The topic of the study is interesting and significant. There are some issues should be addressed before the publication of the manuscript, and are listed below:

Point 1: The title of the paper is suggested to be changed to “Experimental study on sulfuric acid corrosion resistance of PHC used for pipe pile and NSC commonly used in engineering”

Response 1: Thank you for your advice. We have modified the title as follow:“Experimental study on sulfuric acid corrosion resistance of PHC used for pipe pile and NSC commonly used in engineering”.

Point 2: The first paragraph of the introduction is too long, and it is recommended that it be divided into subparagraphs.

Response 2: Thank you for your advice. We have divided the first paragraph of the introduction into three subparagraphs.

Point 3: Some recently literatures are suggested to add and refer in the present paper. Such as " Permeability evolution model of coarse porous concrete under sulphuric acid corrosion. Construction and Building Materials, 326,2022, 126475; Corrosion of steel rebars across UHPC joint interface under chloride attack. Construction and Building Materials, 2023, 387: 131591; Experimental Study on Bond Behavior between Steel Rebar and PVA Fiber-Reinforced Concrete. Coatings, 2023, 13(4): 740. Corrosion-Effected Bond Behavior between PVA-Fiber Reinforced Concrete and Steel Rebar under Chloride Environment. Materials, 2023, 16(7): 02666;

Response 3: Thank you for your advice. We have added and referred the papers.

Point 4: The mixture proportion of C30, C50 and C80 designed in the present test should be added.

Response 4: Thank you for your advice. We have added the mixture proportion of C30, C50 and C80 designed in the present test.

Local Portland cement (42.5 P â…¡ ) produced by China Resources Cement Holdings Limited was used, which complies with Chinese standard GB175-2007. Mixture proportions for concrete specimens are given in Table 1. Naphthalene based superplasticizer was added to obtain sufficient workability. The production of pipe piles is mainly achieved through high-speed centrifugal molding of molds, which generates a large amount of residual slurry during the molding process. In addition to water, the main components of the residual slurry produced by pipe pile production are cement, ground sand, a small amount of fine sand, and very little water reducing agent. The residual slurry contains about 70% liquid and 30% solid. After years of testing, the pipe pile enterprise has obtained a surplus slurry content that meets the performance requirements, and has achieved good economic and environmental protection benefits. The specific surface area of the ground sand used is greater than 420m2/kg, and the silicon dioxide content is greater than 90%. When the ground sand is added to cement concrete, it can further react with the silicon dioxide in the ground sand and the calcium hydroxide in the hydration product of cement concrete under autoclave curing conditions to generate tobermorite with high strength, good crystallinity, and stability.

Table1  Mixture proportions of C30, C50 and C80 concretes(kg/m3)

Strength grade

Cement

Fly ash

Mineral powder

Ground sand

Sand

Gravel

Superplasticizer

Residual slurry

Water

C30

198

66

66

/

780

1075

10.8

/

155.0

C50

255

/

/

135

750

1300

9.5

150

/

C80

255

/

/

135

720

1330

9.5

180

/

Point 5: And, the material properties of the aggregates should also be added.

Response 5:  Thank you for your advice. We have added the material properties of the aggregates in the present test. Granite gravels with a maximum nominal size of 31 mm were obtained for concrete specimens. Table 2 shows the particle size gradation of gravel (coarse aggregates) used in the trial mixture.

Table 2 Results of sieving test for Granite gravel

Sieve size (mm)

31.5

26.5

19.0

16.0

9.5

4.75

2.36

<2.36

Grader retained (%)

0

1.5

43.5

6.4

36.4

2.7

4.1

5.4

Accumulated retained(%)

0

2

45

51

88

91

95

100

Point 6: The size of each specimen in Figure 3 is recommended to be adjusted to the same.

Response 6: Thank you for your advice. We have adjusted Figure 3.

Point 7: The summary in lines 282-283 “the higher the strength grade, the greater the mass loss of the specimen” is inaccurate. As indicated that the sulfuric acid corrosion does not cause the mass loss in specimens with C30 and C50 in the present test.

Response 7: Thank you for your advice. We have adjusted the sentence to be “the higher the strength grade, the greater the mass change of the specimen”.

Point 8: Table 1 shows that 3 replicates were set up for each experimental condition, but only one curve and one peak stress are given in Figures 6 and 7 for each experimental condition, are they average values? How to give the average of the three load-slip curves? Please explain.

Response 8: Thank you for your advice. This is a very important question. For the full stress-strain curve in Figure 6, we selected the curve whose strength was the middle value in a group of three specimens, while the peak stress in Figure 7 we selected the average value of the peak stress in a group of three specimens.

Point 9:  Lines 386-388 of the conclusion are an exact duplicate of lines 389-391.

Response 9: Thank you for your advice. Sorry, we made a slip of the pen that caused the same sentence to appear twice

Point 10: Some quantitative results are suggested in lines 392-399 of the conclusion.

Response 10: Thank you for your advice. We have added Some quantitative results as follows: “After 168 days of accelerated corrosion in sulfuric acid solution, the peak stress of C30, C50, and C80 specimens decreased by 34.97%, 36.76%, and 28.27%, respectively.”

Reviewer 3 Report

The manuscript entitled, "Experimental study on sulfuric acid corrosion resistance of concrete for PHC pipe pile and two kinds of concrete commonly used in engineering" reports on the change in mechanical properties of different types of concrete exposed to sulfuric acid immersion as part of an accelerated corrosion test.  I found the article to be very interesting and informative and I think it is relevant to the journal's aims and will be of interest to its readers.

I wonder if the authors could expand their discussion on the reasons for the decrease in peak stress of the C30 concrete? The explanation for the decreases in the C50 and C80 concretes are consistent with the penetration of the corrosion attack into the sample and the formation of corrosion products with a larger volume that then cracks the concrete.  However, in the C30 samples, the corrosion depth increases due to the formation of corrosion products outside the cylindrical sample as well as, presumably, some corrosion products in the interior.  However, the articles states that there is greater porosity in the C30 concretes so is the higher porosity sufficient to allow for the formation of corrosion products?  Clearly, the experiments show a decrease in the mechanical properties of the C30 specimens, so I'm wondering if the sulfuric acid neutralization reaction could also be removing the "gluing" effect of the cement between the aggregates as a contributor to the loss of mechanical properties?

I saw only a couple of minor grammatical concerns.  Very well written!

Author Response

Response to Reviewer 3 Comments

Point 1:The manuscript entitled, "Experimental study on sulfuric acid corrosion resistance of concrete for PHC pipe pile and two kinds of concrete commonly used in engineering" reports on the change in mechanical properties of different types of concrete exposed to sulfuric acid immersion as part of an accelerated corrosion test.  I found the article to be very interesting and informative and I think it is relevant to the journal's aims and will be of interest to its readers.

I wonder if the authors could expand their discussion on the reasons for the decrease in peak stress of the C30 concrete? The explanation for the decreases in the C50 and C80 concretes are consistent with the penetration of the corrosion attack into the sample and the formation of corrosion products with a larger volume that then cracks the concrete.  However, in the C30 samples, the corrosion depth increases due to the formation of corrosion products outside the cylindrical sample as well as, presumably, some corrosion products in the interior.  However, the articles states that there is greater porosity in the C30 concretes so is the higher porosity sufficient to allow for the formation of corrosion products?  Clearly, the experiments show a decrease in the mechanical properties of the C30 specimens, so I'm wondering if the sulfuric acid neutralization reaction could also be removing the "gluing" effect of the cement between the aggregates as a contributor to the loss of mechanical properties?

Response 1:Thank you for your advice. This is a very important question.

Liu drew a schematic diagram of the comparison of concrete before and after corrosion in sulfate environment based on a large number of corrosion test results. Due to the diffusion effect of corrosive media, concrete specimens undergo corrosion from the surface to the inside. The light colored area in Figure 1 (b) represents the uncorroded concrete, which is called the uncorroded area, while the dark colored area represents the concrete corroded by corrosive media, which is called the corroded area. The degradation of concrete mechanical properties is mainly caused by the degradation of concrete mechanical properties in the dark corrosion zone.

“Liu H, Li J. Constitutive Law of Attacked Concrete[J]. Journal of Building Materials, 2011, 14(6): 736-741.(in Chinese)”

(a) Uncorroded cross section               (b) Partly corroded cross section

Kawai found that concrete with a high water cement ratio has larger and more pores than that with a low water cement ratio. These pores play the role of a capacity to absorb expansion caused by the production of gypsum. Therefore concrete with a high water cement ratio has a higher capacity to absorb the expansion of production reaction of gypsum than that with a low water cement ratio, and this phenomenon is schematically illustrated in Fig. 8. (Kawai K, Yamaji S, Shinmi T. Concrete Deterioration Caused by Sulfuric Acid Attack[C]. 10th International Conference on Durability of Building Materials and Components, 2005.)

Figure 8. Mechanism of concrete deterioration due to sulfuric acid attack

Reviewer 4 Report

This study deals with the sulfuric acid corrosion resistance of concrete for PHC pipe pile and two types of concrete that are commonly used. The paper is well organized, and in the introduction the scope is made clear. Conclusions are also clear. I suggest that the paper can be accepted with minor changes.

-      -  The title needs to be improved. “Two kinds of concrete” can be better phrased.

-       - You could give some more details of the sulfuric acid solution except pH.

-      -  Give more details about aggregates (gravel and sand). Was it quartz or limestone sand? What was the size distribution? Since in the introduction you say that sulfiric resistance of concrete depends on cement content among other factors, why don’t you give more details about the mix design.

-     -   In Figures 4,5,7,8,9 it is preferred not to use lines if you do not know the relationship.

-       - What is the innovation in this study? Describe it clearly in introduction.

Author Response

Response to Reviewer 4 Comments

This study deals with the sulfuric acid corrosion resistance of concrete for PHC pipe pile and two types of concrete that are commonly used. The paper is well organized, and in the introduction the scope is made clear. Conclusions are also clear. I suggest that the paper can be accepted with minor changes.

Point 1: The title needs to be improved. “Two kinds of concrete” can be better phrased.

Response 1: Thank you for your advice. We have modified the title as follow:“Experimental study on sulfuric acid corrosion resistance of PHC used for pipe pile and NSC commonly used in engineering”.

Point 2: You could give some more details of the sulfuric acid solution except pH.

Response 2: Thank you for your advice. We have added A portable pH meter (0.01 precision) was used to measure pH of solutions during the testing period of 165 days. It was kept in the range of 0.83–0.87 by daily adjusting the pH value using concentrated sulfuric acid (98% by weight). The solution was thoroughly stirred after adding the concentrated sulfuric acid everyday in order to reduce differential concentrations of the acid within the solution vessel.”

Point 3:  Give more details about aggregates (gravel and sand). Was it quartz or limestone sand? What was the size distribution? Since in the introduction you say that sulfiric resistance of concrete depends on cement content among other factors, why don’t you give more details about the mix design.

Response 3: Thank you for your advice. We have added the details about aggregates and mix design as follows:

Local Portland cement (42.5 P â…¡ ) produced by China Resources Cement Holdings Limited was used, which complies with Chinese standard GB175-2007.  Granite gravel was used as coarse aggregates and machine-made sand was used as fine aggregates. In order to represent the concrete material used for PHC pipe piles, the mix ratio of C80 concrete specimens was consistent with that of a PHC pipe pile plant in Guangdong province, where the forming and curing of specimens were also carried out. The C50 and C30 concrete specimens were poured at a mixing plant near the PHC pipe pile plant in order to represent the concrete commonly used in engineering. Mixture proportions for concrete specimens are given in Table 1. Naphthalene based superplasticizer was added to obtain sufficient workability. The production of pipe piles is mainly achieved through high-speed centrifugal molding of molds, which generates a large amount of residual slurry during the molding process. In addition to water, the main components of the residual slurry produced by pipe pile production are cement, ground sand, a small amount of fine sand, and very little water reducing agent. The residual slurry contains about 70% liquid and 30% solid. After years of testing, the pipe pile enterprise has obtained a surplus slurry content that meets the performance requirements, and has achieved good economic and environmental protection benefits. The specific surface area of the ground sand used is greater than 420m2/kg, and the silicon dioxide content is greater than 90%. When the ground sand is added to cement concrete, it can further react with the silicon dioxide in the ground sand and the calcium hydroxide in the hydration product of cement concrete under autoclave curing conditions to generate tobermorite with high strength, good crystallinity, and stability.

Table 1 Mixture proportions of C30, C50 and C80 concretes(kg/m3)

Strength grade

Cement

Fly ash

Mineral powder

Ground sand

Sand

Gravel

Superplasticizer

Residual slurry

Water

C30

198

66

66

/

780

1075

10.8

/

155.0

C50

255

/

/

135

750

1300

9.5

150

/

C80

255

/

/

135

720

1330

9.5

180

/

Granite gravels with a maximum nominal size of 31 mm were obtained for concrete specimens. Table 2 shows the particle size gradation of gravel (coarse aggregates) used in the trial mixture.

Table 2 Results of sieving test for Granite gravel

Sieve size (mm)

31.5

26.5

19.0

16.0

9.5

4.75

2.36

<2.36

Grader retained (%)

0

1.5

43.5

6.4

36.4

2.7

4.1

5.4

Accumulated retained(%)

0

2

45

51

88

91

95

100

Point 4:  In Figures 4,5,7,8,9 it is preferred not to use lines if you do not know the relationship.

Response 4: Thank you for your advice. The question you raised is very good. Indeed, we have only obtained data points through experiments, and we should not use lines to connect the data points. However, we still think that using lines to connect the data points is more beautiful, so we are not sure if we can maintain the original way of drawing points. Please understand.

Point 5: What is the innovation in this study? Describe it clearly in introduction.

Response 5: Thank you for your advice. the purpose and innovation of this study:The concealment of PHC pipe piles is easy to cause people's neglect, and more attention should be paid to the durability design of PHC pipe piles [37-39]. Based on this, in this paper, concrete samples with three different grade strengths of C30, C50 and C80 were selected, and sulfuric acid solution with a pH of 0.85 was used as the corrosive medium to conduct an accelerated corrosion test on concrete. The time-varying law of appearance, mass loss, corrosion depth and mechanical prop-erties under uniaxial compression of PHC pipe pile concrete (represented with C80) and two kinds of concrete commonly used in engineering (represented with C30 and C50) samples in the sulfuric acid environment is studied. The research results would provide references for the durability design of concrete structures and the prediction of concrete service life in a sulfuric acid environment.

Round 2

Reviewer 2 Report

All questions have been clarified and the current status is ready for acceptance

Author Response

Point 1: All questions have been clarified and the current status is ready for acceptance

Response 1: Thank you from the bottom of my heart for your hard work and dedication.

Reviewer 4 Report

Beauty is subjective, science is objective. Since you agree that you should not use the lines in the figures, why don't you find a different type of chart or a table.

Line 131: two "pile" typing

Author Response

Point 1: Beauty is subjective, science is objective. Since you agree that you should not use the lines in the figures, why don't you find a different type of chart or a table.

Response 1: Thank you for your advice. We have revised Figures 4,5,7,8,9 as follow,and added a sentence “It should be noted that although it is easier to see the trend of changes by connecting scattered points into a line, it is not scientific enough because we only obtained data at the scattered points in our experiment. Therefore, in this article, we use the form of drawing scattered points to reflect the experimental results.  ”

Figure 4 Corrosion depth changes with corrosion time

Figure 5 The percentage of mass loss changes with corrosion time

Figure 7 Variation of peak stress with corrosion time

Figure 9 Variation rate of compressive strength with corrosion age

Figure 11 Variation of elastic modulus with corrosion age

Line 131: two "pile" typing
